# Molecular insights into the catalytic mechanism of plasticizer degradation by a monoalkyl phthalate hydrolase

Yebao Chen[1], Yongjin Wang[2], Yang Xu[3], Jiaojiao Sun[3], Liu Yang[3], Chenhao Feng[3], Jia Wang[3], Yang Zhou ⓘ [2,5✉], Zhi-Min Zhang ⓘ [2,5✉] & Yonghua Wang[3,4,5✉]

Phthalate acid esters (PAEs), a group of xenobiotic compounds used extensively as plasticizers, have attracted increasing concern for adverse effects to human health and the environment. Microbial degradation relying on PAE hydrolases is a promising treatment. However, only a limited number of PAE hydrolases were characterized to date. Here we report the structures of MehpH, a monoalkyl phthalate (MBP) hydrolase that catalyzes the reaction of MBP to phthalic acid and the corresponding alcohol, in apo and ligand-bound form. The structures reveal a positively-charged catalytic center, complementary to the negatively-charged carboxyl group on MBP, and a penetrating tunnel that serves as exit of alcohol. The study provides a first glimpse into the enzyme-substrate binding model for PAE hydrolases, leading strong support to the development of better enzymes in the future.

---

[1] School of Biology and Biological Engineering, South China University of Technology, Guangzhou 510006, China. [2] International Cooperative Laboratory of Traditional Chinese Medicine Modernization and Innovative Drug Development of Chinese Ministry of Education (MOE), College of Pharmacy, Jinan University, Guangzhou 510632, China. [3] School of Food Science and Engineering, South China University of Technology, Guangzhou 510640, China. [4] Guangdong Youmei Institute of Inteligent Bio-manufacturing, Foshan, Guangdong 528200, China. [5]These authors jointly supervised this work: Yang Zhou, Zhi-Min Zhang, Yonghua Wang. ✉email: zhouyang@jnu.edu.cn; 13632107756@163.com; yonghw@scut.edu.cn

Phthalate acid esters (PAEs) are diesters of 1,2-benzendicarboxylic acid that have been utilized in plastics to improve flexibility and durability. Currently, there are nearly 60 different types of PAEs in use for numerous consumer products, including storage containers, food packaging[1], children products[2], medical devices[3], cosmetics[4], electronic products[5] and building materials[6]. As plasticizers, PAEs are not covalently bound to the materials they are added in, meaning they are constantly released into the environment during manufacturing process and from the final products. The widespread use of PAEs makes them a huge burden to the environment, as well a potential threat to human health. The relationship between PAEs exposure and the risk of obesity and endocrine system disorders has been well established by extensive epidemiological studies. Therefore, several phthalate congeners, including dimethyl phthalate (DMP), diethyl phthalate (DEP), dibutyl phthalate (DBP), di (2-ethylhexyl) phthalate (DEHP), di-n-octyl phthalate (DNOP) and benzyl butyl phthalate (BBP), have been listed as priority pollutants[7].

Considering the hazardous nature of PAEs, efficient elimination of them from the environment is of great urgency. A variety of methods have been developed to degrade phthalates via abiotic or biotic ways, such as advanced oxidation process, adsorption, microbial degradation and phytoremediation[8], among which microbial degradation is the most promising strategy[9] for advantages on mass balance-wise[10], high efficiency, environmental safety and low cost[11–13]. Conventional microbial degradation is initiated by step-wise de-esterification reactions on phthalate diester to form phthalate acid (PA), relying on the genes encoding phthalate hydrolases (Fig. 1). PA can be further catalyzed via dioxygenase pathways, and genes involved in PA degradation are usually separated from that of phthalate hydrolases[14]. In the past decades, numerous PAE-degrading bacterial and fungal strains have been isolated and analyzed. Especially, Gram-positive bacteria stand out as the most promising PAE hydrolase yielding microbial species, because of their catalytic potential towards a wide range of PAEs[15].

Compared to the extensive efforts on exploration of PAE-degrading microorganisms, only limited PAE hydrolase genes have been identified and characterized in recent years. To date, there are about thirty bacterial enzymes reported for hydrolysis of structurally diverse PAEs, all of which are esterases containing a conserved catalytic triad (Glu/Asp-His-Ser). Different PAE hydrolases exhibit distinct catalytic activities during hydrolysis of the two ester bonds of PAEs, therefore classified as dialkyl, monoalkyl and dialkyl/monoalkyl PAE hydrolases. Primarily, dialkyl PAE hydrolases convert phthalate diesters to the corresponding monoesters, which can be further degraded to PA by monoalkyl PAE hydrolases. Monoalkyl PAE hydrolases are highly clustered on the phylogenetic tree to form a MEHP hydrolase family, suggestive of a conserved catalytic mechanism. A few PAE hydrolases, such as EstM2[16] and EstJ6[17] from metagenomic sources, are capable of hydrolyzing both phthalate diester and phthalate monoester. Nevertheless, the catalytic mechanisms of these enzymes are barely reported.

MehpH is a monoalkyl phthalate hydrolases discovered from *Gordonia sp*. Strain P8219, a Gram-positive bacteria, and able to dissimilate several different kinds of monoester, such as MBP, MEP and MEHP[18]. Here, we report the crystal structure of MehpH both in apo form and in complex with PA and butanol, providing a glimpse into the precise enzyme-substrate binding mode and the catalytic process. This study not only facilitates our understanding about the catalytic mechanism of the hydrolase, but also provides the basis for further enzymatic modification.

## Results

**The structure of apo-MehpH reveals a penetrating tunnel adjacent to the active site.** To reveal the catalytic mechanism of phthalate hydrolases, we first solved the structure of MehpH. The final construct used for crystallization contained a deletion of 23 amino acids on the N-terminus, which are predicted to be disordered. The structure of MehpH, determined at a resolution of 2.3 Å (Table 1 and Supplementary Data 1), exhibits the typical fold of the α/β hydrolase superfamily and is composed of an α/β hydrolase core (residues 24-152 and 225-311) and an α-helical lid domain (residues 153-224) (Fig. 2a). The twist β-sheet in the center of the α/β hydrolase core is formed by eight mixed β-strands (β1-β8) and surround by nine α-helices. Some residues in the the NC-loop (residues 153-164) that refers to a protein segment of variable length and structure linking β6 to the first structurally conserved helix in the lid domain were not assigned because of the discontinuity of the electron density. There is a penetrating tunnel formed by Loop$_{β3αA}$, αA, Loop$_{β5αD}$, Loop$_{β8αK}$ and αK from the α/β hydrolase core and the NC-loop, αE, αF-αG from the lid domain (Fig. 2b). The tunnel is narrow in the body of the structure, but significantly enlarged at the substrate entrance, due to the flexibility of the NC-loop. The catalytic triad, composed of Ser125, Asp259

**Fig. 1 Degradation steps of PAEs by PAE hydrolases.** The aliphatic chains of several widely used PAEs are presented.

**Table 1 Data collection and refinement statistics.**

|  | MehpH (PDB Code: 8HGV) | MehpH-ligand (PDB Code: 8HGW) |
|---|---|---|
| **Data collection** | | |
| Space group | P 2₁ 2₁ 2₁ | I1 2 1 |
| Cell dimensions | | |
| $\quad a, b, c$ (Å) | 77.69, 128.12, 54.46 | 77.0, 49.0, 306.3 |
| $\quad \alpha, \beta, \gamma$ (°) | 90.00, 90.00, 90.00 | 90.00, 93.73, 90.00 |
| Resolution (Å) | 22.13-2.30 | 46.38-2.80 |
|  | (2.39-2.30)ᵃ | (2.97-2.80) |
| $R_{merge}$ | 0.041 (0.235) | 0.365 (0.995) |
| $I/\sigma(I)$ | 7.0 (1.5) | 8.2 (3.8) |
| $CC_{1/2}$ | 0.995 (0.752) | 0.950 (0.826) |
| Completeness (%) | 99.9 (100.0) | 99.9 (100.0) |
| Redundancy | 1.9 (2.0) | 6.4 (6.6) |
| **Refinement** | | |
| Resolution (Å) | 20.74-2.30 | 44.1-2.80 |
| No. reflections | 24809 | 28611 |
| $R_{work}$ / $R_{free}$ | 0.196/0.237 | 0.228/0.260 |
| No. atoms | | |
| $\quad$ Protein | 4268 | 8621 |
| $\quad$ Ligand | 0 | 53 |
| $\quad$ Water | 73 | 0 |
| $B$ factors | | |
| $\quad$ Protein | 60.2 | 53.5 |
| $\quad$ Ligand | - | 45.4 |
| $\quad$ Water | 55.0 | - |
| R.m.s.deviations | | |
| $\quad$ Bond lengths (Å) | 0.003 | 0.005 |
| $\quad$ Bond angles (°) | 0.71 | 1.00 |

ᵃValues in parentheses are for highest-resolution shell.

and His291 with His291 bridging Asp259 and Ser125 through hydrogen bonds, is situated at the bottle-neck of the tunnel, suggesting this tunnel may be involved in the hydrolysis process of MehpH.

A Dali structural homology search revealed that MehpH shares the highest similarity with some meta-cleavage product hydrolases, with the Z-scores above 28.0 and root-mean-square deviation (RMSD) of 2.3–2.8 Å. The best matched one is DxnB2 (PDB code 4LXH, Z-score 29.9, RMSD 2.4 Å), which catalyzes C–C bond hydrolysis of the recalcitrant metabolites of polychlorinated biphenyls, a group of highly carcinogenic compounds formerly used in industry and consumer products. Structure overlap exhibited that MehpH mainly shares high similarity with DxnB2 in the α/β hydrolase core, while their lid domains vary obviously (Supplementary Fig. 1), further supporting the ability of the α/β hydrolase core to accommodate different lid domains to achieve diverse substrate and chemical specificities.

**The structure of MehpH-ligand complex captured a covalently-bound PA.** Next, we investigated the mechanism underlying substrate binding of MehpH by solving the structure of MehpH-MBP complex. A catalytically inactive nucleophile mutation D259N was introduced in the active site to capture the transition state of MehpH-MBP interaction (Supplementary Fig. 2), and the subsequent structure was finally determined at a resolution of 2.8 Å (Table 1 and Supplementary Data 2). Structure superposition of the apo and ligand-bound MehpH revealed no significant conformational change on the whole structure of MehpH, except for the stabilization of the NC-loop in the lid domain and formation of a narrow hole extending from the protein surface to the catalytic Ser125 (Fig. 2c, d). Surprisingly, instead of a separate electron density for MBP in the catalytic center, we observed extra electron density on the side chain of Ser125, indicative of an

intermediate trapped in the structure (Supplementary Fig. 3). The shape of the electron density is consistent with that of PA, suggesting that Ser125 is covalent modified by PA. Additionally, adjacent to the catalytic center, we discovered the electron density of butanol in the penetrating tunnel. It appears that the butanol is highly dynamic in the structure, as only one butanol molecule with intact electron density in the four MehpH of a single asymmetric unit. It is possible that the D259N mutant may retain week hydrolysis activity, so MBP was hydrolyzed during protein crystallization. Together, we captured one of the intermediate states during the reaction, in which the PA is still covalently linked to the catalytic Ser125 through an ester bond and waiting for hydrolysis by a water molecule.

**The active site of MehpH.** PA is tightly embedded in a positively charged pocket that is made up of residues His53, Phe56, Ser125, Arg126, Thr152 and His291 from the α/β hydrolase core and Asp162, Met163 and Tyr166 from the lid domain (Fig. 3a). Particularly, Arg126 and Asp162 construct a gate to lock PA in the pocket through two hydrogen bonds formed by their side chains (Supplementary Fig. 3b). The benzyl ring of PA is sandwiched between Arg126 by a cation-π interaction and on the opposing face Tyr166 through van der Waals' interactions. The backbone NH groups of Gly52 and Arg126 that constitute an oxyanion hole in the catalytic center, a common feature in serine hydrolase such as esterase and lipases[19], are within hydrogen bond distance to one of the O atoms on the carbonyl group that forms an ester bond with the catalytic Ser125 (Fig. 3b). The other carbonyl group on PA is anchored by hydrogen bonds contributed by the backbone NH groups of Ala151 and Thr152, and the side chain of Thr152 (Fig. 3c). In contrast to the extensive interactions of PA with MehpH, the butanol is only engaged one hydrogen bond with Tyr166 (Fig. 3a). Of note, the location of the butanol is very close to His291 of the triad, leaving no space for a water molecule to attach the Ser125-PA intermediate.

Based on the above structural analysis, additional variants were constructed and their MBP hydrolyzing activity was measured (Fig. 3d). First, substitution of Arg126 with an alanine almost abolished the activity (2.66 ± 0.88%), consistent with the observations that Arg126 plays multiple roles in direct recognition of substrate. To further evaluate the importance of Arg126, we replaced it with amino acids containing different types of side chain. All the subsequent mutants, including R126K, R126H, R126Y, R126T, R126N, R126Q, R126D, R126E and R126L, showed obviously reduced activity (Supplementary Fig. 4). Another notable mutation occurs on Thr152, which is supposed to bind the carboxyl group on MBP. The resulting 9.12 ± 1.53% relative activity of T152A indicated the indispensable role of Thr152-MBP interaction during the hydrolysis process. In addition, other PA-interacting residues including Phe56 and Tyr166 are also essential, as each corresponding Ala variant showed significantly lower activity. The two residues on the NC-loop, Asp162 and Met163, appears to play supportive role in hydrolyzing MBP, as D162A and M163A only exhibited moderately compromised activity (41.05 ± 3.55 and 71.72 ± 2.50).

**The tunnel serves as exit for the butanol.** Proteins in the the α/β-hydrolases fold family exhibit a diverse tunnel network in their structures. These tunnels connect the buried active site with the environment, providing access of the smaller reaction components, such as water, to the active site[20]. The observation of butanol sitting in a deeper position of the penetrating tunnel promotes us to hypothesis that it may leave the enzyme through this active site back entrance. We first investigated the tunnel network in MehpH and identified two new tunnels (T2 and T3)

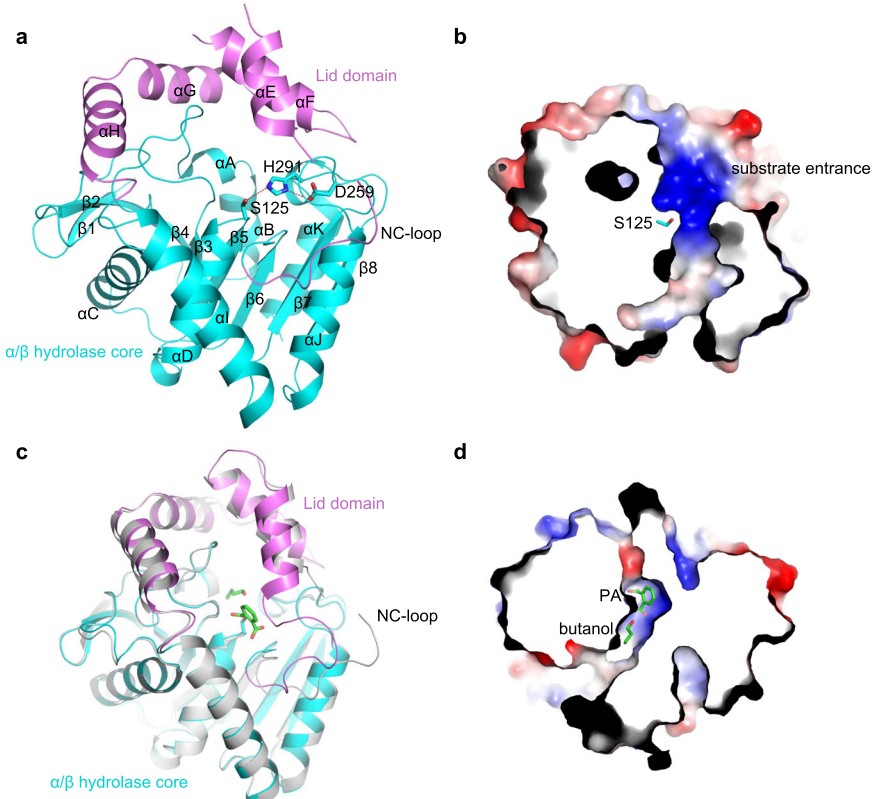

**Fig. 2 Structure of MehpH in apo and in complex with ligand. a** Crystal structure of MehpH in apo form. The lid domain and α/β hydrolase core are colored in purple and cyan, respectively. Similar colors are used in other panels unless otherwise indicated. The catalytic triad (Ser125, Asp259 and His291) is shown in stick representation. The α-helices and β-strands are labeled from A to K and from 1 to 8, respectively. **b** The penetrating tunnel in the structure of apo-MehpH. **c** Structure superposition of apo-MehpH and MehpH-ligand complex, in which apo-MehpH is colored in grey. The Ser125-PA covalent intermediate and butanol are shown as sticks. **d** The location of Ser125-PA covalent intermediate and butanol in the penetrating tunnel of MehpH-ligand complex.

besides the one found in the structure (T1) (Fig. 4a). These three tunnels merge at the butanol binding site. To further determine the egress tunnel of butanol, we carried out 24 replicas of unbiased molecular dynamics (MD) simulations starting from the butanol-bound state. The simulation time for each replica was set to 200 ns, resulting a total simulations time of 4.8 μs. As shown from the root-mean-square deviation (RMSD) of butanol, 7 egressing events were observed during the simulation (Fig. 4b). The trajectories of these events reveal that butanol left the active site mainly through T1 (Fig. 4c and Supplementary Fig. 5). The trajectories were projected to a 2D population density map (PDM) using path collective variables (see method section), with $S_{path}$ describing the progression along and $Z_{path}$ describing the distance from the path. The PDM revealed two metastable states (M1 and M2) that most of the unbinding process of butanol went through (Fig. 4d, Supplementary Data 3 and 4). Four bulky residues, including Tyr184, His185, Tyr199, and His292, were found lining the tunnel T1 (Fig. 4e). To support the molecular dynamic studies, we selected two residues (His185 and His292) and replaced them with Arg, respectively. The reason we chose Arg is that its side chain is long enough to block the tunnel, while very flexible to avoid introducing steric conflicts with surrounding residues. As expected, both variants exhibited marked lower activity (Fig. 4f).

## Discussion

In recent years, enzymatic depolymerization of plastics has emerged as an alternative technology to conventional mechanical

and chemical plastic recycling approaches. Huge success has been achieved on the charicaterization and mechanistic study of several enzymes that depolymerize polyethylene terephthalate (PET)[21–24]. However, in addition to the production of monomers that form the plastics, PAEs are also released during the polymer hydrolysis process as they are not covalently bound in the plastics. While this paper was under preparation, von Haugwitz et al. reported the structural study of TfCa, a promiscuous carboxylesterase from *Thermobifida fusca* that could hydrolyze PET degradation intermediates such as bis(2-hydroxyethyl) terephthalate (BHET) and mono-(2-hydroxyethyl)-terephthalate (MHET) to generate terephthalate (TPA). They also investigate the catalytic activity of TfCa to DEP (Fig. 1), and found that TfCa completely converted DEP to MEP, but cannot further hydrolyze MEP to PA[24], highlighting the significance of developing PAE hydrolases.

One interesting question is whether or not PAE hydrolases could degrade terephthalate-based mono- and di-esters generated during PET degradation. Structural comparison of MehpH with TfCa and MHETases, another well-known PET-degrading enzyme[21,23], reveals similar orientations of ligands bound in the active site (Supplementary Fig. 6). However, both TfCa and MHETase have spacious cavities to accommodate the carboxyl group or its esterified derivative on the tere-position of phthalate. While in MehpH, this site is occupied by the NC-loop, suggesting future engineering work on the NC-loop may convert MehpH to a terephthalate-based esterase.

PAE hydrolases are attracting significant attention for their wide distribution in the nature and hydrolysis activities against structurally diverse PAEs. Nevertheless, the underlying mechanisms on

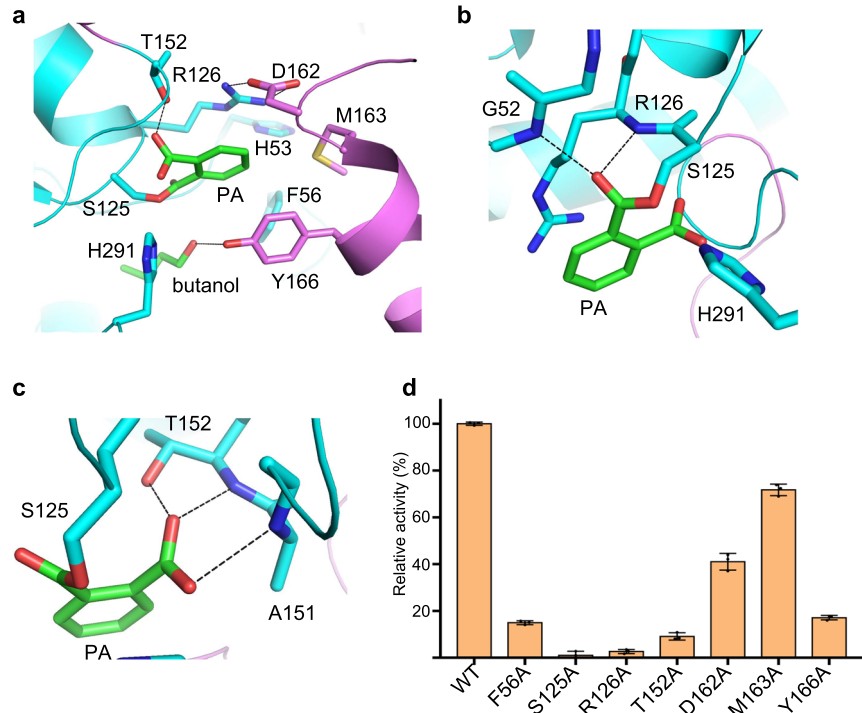

**Fig. 3 The substrate binding pocket of MehpH. a** Residues involved in the active site of MehpH. The hydrogen bonds formed between the residues and the substrate are shown as black-colored lines. **b** The oxyanion hole in the active site. **c** The interactions between MehpH and the free carboxyl group on PA. **d** Hydrolytic activities of MehpH and its variants using MBP as substrate. The experiments were carried out using MBP concentration of 1.0 mM and enzyme concentration of 1.0 μM. The amount of MBP and produced PA was monitored by HPLC at a wavelength of 254 nm. The activity of wild type (WT) MehpH was set as 100%, and the hydrolytic activities of the variants were compared to that of the WT. Error bars represent the s.d. values obtained in triplicate experiments.

the exceptional abilities of different phthalate hydrolases remain unclear. In this work, the crystal structures of MehpH, a monoalkyl phthalate hydrolase, in apo and in complex with PA and butanol provide a first glimpse into the precise enzyme-substrate binding mode and the catalytic process. It appears that the NC-loop serves as a gatekeeper of the enzyme. In the absence of MBP, the NC-loop becomes disordered to keep the the active site open. When MBP enters the active site, the NC-loop moves back to block the entrance. At the same time, a set of residues lining the binding pocket recognize both the benzyl ring and the carboxyl group of MBP simultaneously, thus positioning the fatty chain into a penetrating tunnel and the target ester bond close to the catalytic Ser125. Ser125, deprotonated by His291, performs nucleophilic attack at the carbonyl group in the ester bond, resulting departure of the produced alcohol via the penetrating tunnel and formation of a Ser125-PA covalent intermediate. The intermediate should be hydrolyzed by a water molecule to generate PA, which leaves the active site accompanying the dislocation of the NC-loop.

## Methods

**Strains, plasmids and chemicals**. The strain *E. coli* BL21(DE3) was obtained from Novagen. The plasmid *pET30a-MehpH* was synthesized by Sangon (Shanghai, China). Oligonucleotides for cloning were synthesized by InvitrogenTM (Thermo Fisher Scientific (China) Co., Ltd. Shanghai, China. Restriction enzymes and DNA polymerases were purchased from New England Biolabs (Beverly, MA, USA) or TaKaRa (Dalian, China). Monobutyl phthalate (MBP, purity >98.0%) was supplied by Aladdin Biochemical Technology (Shanghai, China).

**Expression and purification of MehpH**. The *pET30a-MehpH* plasmids were transferred into *E. coli* BL21 (DE3) strain for overexpression. The expression of MehpH was induced by 0.2 mM isopropyl β-d-1-thiogalactopyranoside (IPTG) at 16 °C for 24 h. The cells were harvested by centrifugation, resuspended in buffer A (20 mM NaH$_2$PO$_4$, 20 mM Na$_2$HPO$_4$, 500 mM NaCl and 30 mM imidazole), and then lysed by sonication. His-tagged MehpH were purified with a HiTap

column and a Superdex-200 16/600 column pre-equilibrated with buffer B (10 mM Tris-HCl, 150 mM NaCl, pH 8.0). Purified MehpH proteins were stored at −80 °C for future use.

**Structure determination of MehpH**. MehpH was crystallized using the handing-drop vapor diffusion method at 16 °C by mixing 1.0 μL MehpH (8.0 mg/mL) with equal volumes of mother liquor containing 100 mM Tris, pH 7.2, 17% (w/v) PEG 3350 and 200 mM Calcium Acetate. For MehpH-MBP complex, crystal growed at the condition of 100 mM Bis-Tris, pH 6.2, 22% (w/v) PEG 3350 and 250 mM MgCl$_2$. Before harvest, crystals were cryo-protected in mother liquor supplemented with glycerol to a final concentration of 25% (v/v) and rapidly cooled into liquid nitrogen. X-ray diffraction data were collected at the 18U1 beamline of the Shanghai Synchrotron Radiation Facility (SSRF, Shanghai, China). Image indexing, integration, and data scaling performed with the XDS package[25,26] and the CCP4 suite[27]. The structures were solved by molecular replacement using alphafold2-generated model. Iterative cycles of model rebuilding and refinement were carried out with COOT and PHENIX[28,29].

**In vitro enzyme assays**. MBP hydrolysis reactions were performed in 5 mL glass vial with sealed lid in a water bath. The reaction mixture (1.0 mL) was consisted of 100 μL MBP solution (10 mM in DMSO) and 900 μL enzyme solution. 100 μL HCl (3.0 M) was added to terminate the reaction at the specific time, after which ethyl acetate (1.0 mL) was used to extract the reaction mixture. The buffer without enzyme was used as negative control. Before HPLC analysis, all samples dissolved in ethyl acetate were filtered with a 0.22 μm filter membrane. The concentration of product and residual substrate were determined by high-performance liquid chromatography (HPLC) using Waters e2695 system (Waters Corporation, Massachusetts, USA) equipped with a 2489 UV/Vis detector and a SunFire C18 column (4.6 mm × 250 mm, 5 μm). The detection wavelength and column temperature were set to 254 nm and 30 °C, respectively. The mobile phase were methanol/water/formic acid (A, 90:10:1, vol/vol/vol) and water. The gradient elution procedure was set as: 80% A (0 min), 100% A (5–10 min), 80% A (12–15 min) at a flow rate of 1.0 mL/min. The retention time of MBP and PA were 5.05 and 3.79 min, respectively.

**Computational simulations**. The tunnels in MehpH were predicted by CAVER 3.0[30] with default settings (the "average bottleneck radius" is 2.01 Å). Molecular dynamics simulations were carried out using Desmond[31]. The structure was

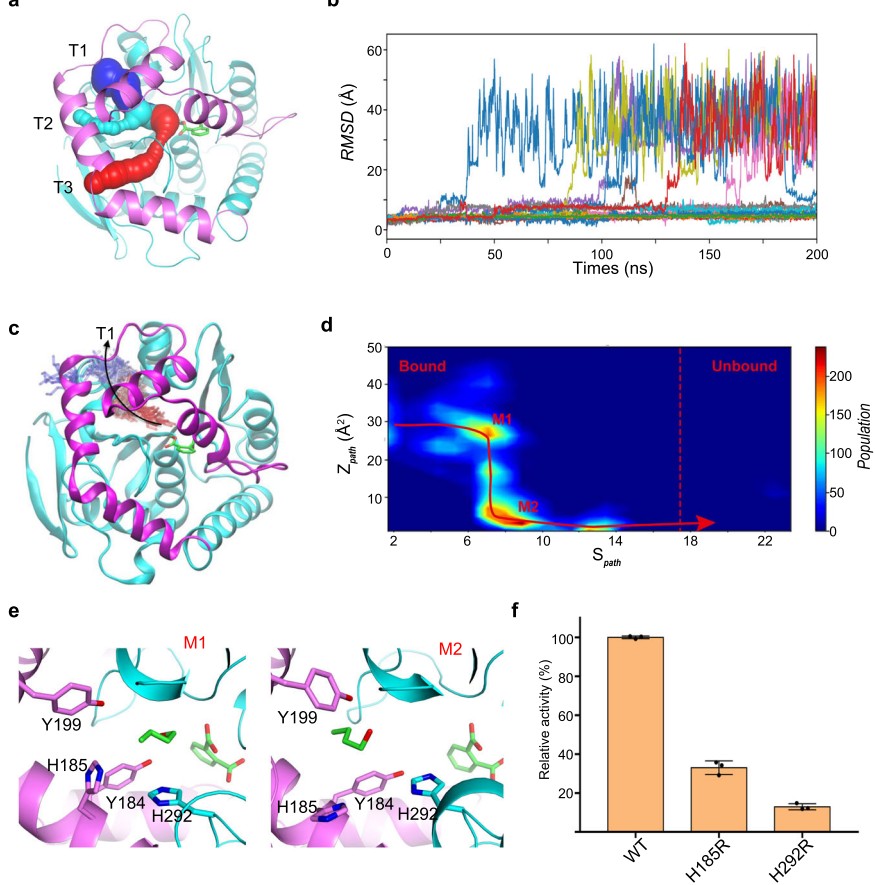

**Fig. 4 Egress tunnel analysis for butanol. a** The predicted tunnel of MehpH using CAVER 3.0. **b** The RMSD values of butanol from 24 independent MD simulation runs. **c** A representative unbinding process of butanol: time evolution of butanol is colored from red (t = 0) to blue (unbound). **d** Population density map for butanol unbinding from the active center. **e** The representative structures for metastable state M1 and M2. **f** Hydrolytic activities of MehpH variants with mutations in the penetrating tunnel. Error bars represent the s.d. values obtained in triplicate experiments.

prepared using the Protein Preparation Wizard (Schrödinger, LLC, New York, NY, 2021). The OPLS4 force filed was used for the substrate and the enzyme and SPC solvent model used to solvate the system. The system was subjected to energy minimization and relaxed in NVT, NPT ensemble before 200 ns production runs at 300 K and 1 atm. The obtained trajectories were analyzed using path collective variables introduced by Branduardi et al.[32]. The progression along the reference path ($S_{path}$) and the deviation from the reference path ($Z_{path}$) are defined as

$$S_{path} = \frac{\sum_{i=1}^{n} i exp(-\lambda msd(X, X_i))}{\sum_{i=1}^{n} exp(-\lambda msd(X, X_i))}$$

and

$$Z_{path} = \frac{\ln(\sum_{i=1}^{n} exp(-\lambda msd(X, X_i)))}{-\lambda}$$

where $X$ denotes the coordinates of interest at the current simulation time-step, $X_i$ denotes the coordinates of the $i$th reference frame that composes the path, $msd(X, X_i)$ is the mean-square deviation between $X$ and $X_i$, and $\lambda$ is a smoothing parameter. One of the unbinding trajectories was used as the reference. The analysis was carried out using VMD[33] and Plumed 2[34].

**Reporting summary**. Further information on research design is available in the Nature Portfolio Reporting Summary linked to this article.

## Data availability
The atomic coordinates of MehpH in apo form (Supplementary Data 1) and in complex with PA and butanol (Supplementary Data 2) have been deposited into the PDB under the accession code 8HGV and 8HGW, respectively. The representative structures for metastable state M1 and M2 generated in Computational Simulations are provided as Supplementary Data 3 and 4, respectively. All other data are available from the Yonghua.W., Z.M.Z. and Y.Z. on reasonable request.

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

## Acknowledgements

This work was supported by funds from National Key R&D Program of China (2022YFC2805100 to Y.W.), National Science Fund for Key Program of National Natural Science Foundation of China (31930084 to Y.W.), National Science Fund for Distinguished Young Scholars of China (31725022 to Y.W.), Natural Science Foundation of China (32171184 to Z.M.Z), Guangdong Province (2019QN01Y979 to Z.M.Z) and Guangdong Basic and Applied Basic Research Foundation (2023A1515012763 and 2022A1515012266 to Z.M.Z). We thank the staffs from BL17B/BL18U1/BL19U1/ BL19U2/BL01B beamline and of National Facility for Protein Science in Shanghai (NFPS) at Shanghai Synchrotron Radiation Facility, for assistance during data collection.

## Author contributions

Y. C. performed most of the experiments, including protein purification, crystallization and enzymatic assays, under the assistance of Y. X., J. S., L. Y. and J. W.. Y. Z. performed computational study. Yongjin W. and C. F. collected and processed the diffraction data for the crystals. Z. M. Z. solved and analyzed the structures. Z. M. Z. and Yonghua W. conceived and oversaw the project.

## Competing interests

The authors declare no competing interests.
