## [Peer Review File · Communications Chemistry]

Reviewers' comments:

Reviewer #1 (Remarks to the Author):

In their manuscript, Chen et al. report the crystal structures a monoalkyl phthalate (MBP) hydrolase that is involved in PAEs biodegradation in apo and MBP-binding form, providing long-awaited details for enzyme-substrate binding in these enzymes. Given the widely concern of PAEs to the environment and human health, this manuscript in principle would be suited to publication in Communications Chemistry.

Having said this, there are several aspects that preclude publication in its present form:

1. There are several other MBP hydrolases reported. It is not discussed in the manuscript how the substrate binding model revealed here is conserved in these enzymes.
2. A catalytically inactive nucleophile mutation D259N was introduced in the active site to capture the transition state of MehpH-MBP interaction. The authors should test the activity of this mutant before crystallization.
3. What was the value of the "average bottleneck radius" calculated by CAVER?
4. Can the authors explain more clearly about the path collective variables (S_{path} and Z_{path}) in Figure 4d?

Reviewer #2 (Remarks to the Author):

The manuscript titled "Molecular insights into the catalytic mechanism of plasticizer degradation by a monoalkyl phthalate hydrolase" reported the crystal structure of an MBP hydrolase MehpH in a ligand-free form and a ligand-bound form with PA and butanol, reflecting a situation after hydrolysis has occurred. Deduced from this ligand-bound structure, the authors performed MD simulations to examine the potential pathway by which butanol as a product can leave the catalytic site and mutagenesis experiments to confirm their hypothesis. Although this paper only covers a relatively narrower scope, the growing interest in enzymes used for environmental remediation suggests that solving additional PAEs hydrolase structures and elucidating their structure-function relationship will undoubtedly contribute to the advancement of future research. Overall, this paper is well-written, and I am confident that it can be published in Communications Chemistry with only minor revisions. I only have a few questions and suggestions that will enhance the quality of the revised manuscript. Major concerns:

1.) The description on page 6 under the subheading "the structure of MehpH-MBP complex captured a covalent intermediate" as well as the description on page 10 in lines 205 to 206 as well as the figure legend for Figure 2, etc. appears to be somewhat confusing. I am aware that the authors attempted to bind an MBP to the structure, but based on Figure 2, it appears to me that the structure was bound to separate PA and butanol, reflecting the situation after the hydrolysis event. Therefore, it is necessary to revise all descriptions of the ligand-bound structure for greater precision and consistency.

2.) Based on a recent study by von Haugwitz et al. (not Haugwitz et al.!) concerning the promiscuous TfCa enzyme, the authors attempted to link the PAEs hydrolases to the PET degrading enzymes. It would be very interesting if the authors included a structural comparison of the MBP hydrolase solved in this study with terephthalate-degrading enzymes such as TfCa and MHETase (a mono-terephthalate ester hydrolyzing enzyme, <https://doi.org/10.1038/s41467-019-09326-3>) to discuss their similarities and differences. In the same vein, it would be very helpful if they could answer the question of whether there is an opportunity to use PAEs hydrolase in the enzymatic plastic degradation process to address the product inhibition challenges caused by terephthalate-based mono- and di-esters, which

have been identified as a key bottleneck for enzymatic PET recycling (<https://doi.org/10.1021/acscatal.1c05856>) and have been mitigated using additional enzymes, such as TfCa (von Haugwitz et al., 2022) and MHETases (<https://doi.org/10.1016/j.checat.2022.11.004>, <https://doi.org/10.1073/pnas.200675311>).

Structure-guided discussion in this direction will undoubtedly attract more scientists focusing on plastic degradation, a highly active area of research in recent years.

One minor issue in the Methods section on page 11, line 227: it is unclear to whom "Macklin" and "Aladdin" refer. In the event that they are chemical suppliers, the authors should provide the full company names and locations, as they did for other commercial suppliers in the same paragraph.

We thank both the reviewers for their positive comments on our work. Please find our point-by-point response to each of the reviewers' comments below.

REVIEWER COMMENTS

Reviewer #1 (Remarks to the Author):

In their manuscript, Chen et al. report the crystal structures a monoalkyl phthalate (MBP) hydrolase that is involved in PAEs biodegradation in apo and MBP-binding form, providing long-awaited details for enzyme-substrate binding in these enzymes. Given the widely concern of PAEs to the environment and human health, this manuscript in principle would be suited to publication in Communications Chemistry.

Response: We are grateful to the reviewer for the positive comments on our manuscript.

Having said this, there are several aspects that preclude publication in its present form:

1. There are several other MBP hydrolases reported. It is not discussed in the manuscript how the substrate binding model revealed here is conserved in these enzymes.

Response: It is reported that Monoalkyl PAEs hydrolases are highly clustered on the phylogenetic tree to form a MEHP hydrolase family and some of the residues involved in substrate binding are known highly conserved, including the catalytic triad. But we also noticed that several reported MBP hydrolases, such as MpeH from *Microbacterium* sp PAE-1, are quite different on the key residues. Since the substrate binding model of these MBP hydrolases is still elusive, we believe it is too early to summarize conserved residues in the substrate binding.

2. A catalytically inactive nucleophile mutation D259N was introduced in the active site to capture the transition state of MehPH-MBP interaction. The authors should test the activity of this mutant before crystallization.

Response: Following the reviewer's suggestion, we tested the *in vitro* activity of MehPH D259N. The result showed that its activity is almost totally abolished. We have added the result in the Supplementary file as Supplementary figure 2.

3. What was the value of the "average bottleneck radius" calculated by CAVER?

Response: The value of the “average bottleneck radius” calculated by CAVER is 2.01 Å. We have added this information in the Methods.

4. Can the authors explain more clearly about the path collective variables (S_{path} and Z_{path}) in Figure 4d?

Response: Following the method proposed by Branduardi et al (*J. Chem. Phys.* 2007, 126, 054103) , we used S_{path} and Z_{path} to describe the position of a point (X_i) with respect to the preassigned path (described by a set of coordinates $X=\{X_1, X_2, \dots, X_n\}$) in the configurational space. According to the definition, S_{path} is a value describing the progression along the preassigned path and Z_{path} represents the distance to the closest point on the path. The path variables are illustrated in the figure below.

(<https://www.plumed.org/doc-v2.3/user-doc/html/belfast-2.html>)

We have added the above explanation into the revised manuscript.

Reviewer #2 (Remarks to the Author):

Although this paper only covers a relatively narrower scope, the growing interest in enzymes used for environmental remediation suggests that solving additional PAEs hydrolase structures and elucidating their structure-function relationship will undoubtedly contribute to the advancement of future research. Overall, this paper is well-written, and I am confident that it can be published in Communications Chemistry with only minor revisions.

Response: We appreciate that the reviewer recognizes the significance of our work. Our responses to the reviewer’s advice are listed below.

Major concerns:

1.) The description on page 6 under the subheading "the structure of MehphH-MBP complex captured a covalent intermediate" as well as the description on page 10 in lines 205 to 206 as well as the figure legend for Figure 2, etc. appears to be somewhat confusing. I am aware that the authors attempted to bind an MBP to the structure, but based on Figure 2, it appears to me that the structure was bound to separate PA and butanol, reflecting the situation after the hydrolysis event. Therefore, it is necessary to revise all descriptions of the ligand-bound structure for greater precision and consistency.

Response: We understand the reviewer's concern. Indeed, it is confusing to use MehphH-MBP complex since MBP has been hydrolyzed in the structure. We have revised the subheading "the structure of MehphH-MBP complex captured a covalent intermediate" to "The structure of MehphH-ligand complex captured a covalently-bound PA". In other parts of the manuscript, the "MehphH-MBP complex" is replaced by either "MehphH-ligand complex" or "MehphH in complex with PA and butanol".

2.) Based on a recent study by von Haugwitz et al. (not Haugwitz et al.!) concerning the promiscuous TfCa enzyme, the authors attempted to link the PAEs hydrolases to the PET degrading enzymes. It would be very interesting if the authors included a structural comparison of the MBP hydrolase solved in this study with terephthalate-degrading enzymes such as TfCa and MHETase (a mono-terephthalate ester hydrolyzing enzyme, <https://doi.org/10.1038/s41467-019-09326-3>) to discuss their similarities and differences. In the same vein, it would be very helpful if they could answer the question of whether there is an opportunity to use PAEs hydrolase in the enzymatic plastic degradation process to address the product inhibition challenges caused by terephthalate-based mono- and di-esters, which have been identified as a key bottleneck for enzymatic PET recycling (<https://doi.org/10.1021/acscatal.1c05856>) and have been mitigated using additional enzymes, such as TfCa (von Haugwitz et al., 2022) and MHETases (<https://doi.org/10.1016/j.checat.2022.11.004>, <https://doi.org/10.1073/pnas.2006753111>). Structure-guided discussion in this direction will undoubtedly attract more scientists focusing on plastic degradation, a highly active area of research in recent years.

Response: We thank the reviewer for pointing out the error in our manuscript and have revised it. Following the reviewer's suggestion, we compared the structure of MehphH with those of TfCa (PDB code 7W1J) and MHETase (PDB code 6JTT). In the active sites, ligands are bound in similar orientations in these structures (as shown in the figure below). However, both TfCa and MHETase have spacious cavities to accommodate the carboxyl

group or its esterified derivative on the tere-position. While in MehphH, this site is occupied by the NC-loop, suggesting that future engineering work on the NC-loop may convert MehphH to a terephthalate-based esterase. We have also added this information in the Discussion.

Comparison of the active sites in MehphH and PET-degrading enzymes.

The structure of MehphH-PA complex is colored in brown. The structure of MHETases in complex with MHET (PDB code: 6JTT) is colored in blue. TfCa-MHETA complex (PDB code: 7W1J) is colored in grey.

One minor issue in the Methods section on page 11, line 227: it is unclear to whom "Macklin" and "Aladdin" refer. In the event that they are chemical suppliers, the authors should provide the full company names and locations, as they did for other commercial suppliers in the same paragraph.

Response: We have provided detailed information for "Aladdin": Aladdin Biochemical Technology (Shanghai, China). We realized that the compound from "Macklin" was actually not used in this study and removed it.

REVIEWERS' COMMENTS:

Reviewer #1 (Remarks to the Author):

This paper could be accepted after revision.

Reviewer #2 (Remarks to the Author):

My concerns were successfully addressed by the authors. In my opinion, the manuscript should be published.